

# Hypoxia-induced factor and its role in liver fibrosis

Jan Mohammad Omar[1], Yang Hai[2] and Shizhu Jin[1]

[1] Department of Gastroenterology and Hepatology, The Second Affiliated Hospital of Harbin Medical, Harbin, Heilongjiang, China
[2] College of International Education, Harbin Medical University, Harbin, Heilongjiang, China

## ABSTRACT

Liver fibrosis develops as a result of severe liver damage and is considered a major clinical concern throughout the world. Many factors are crucial for liver fibrosis progression. While advancements have been made to understand this disease, no effective pharmacological drug and treatment strategies have been established that can effectively prevent liver fibrosis or even could halt the fibrotic process. Most of those advances in curing liver fibrosis have been aimed towards mitigating the causes of fibrosis, including the development of potent antivirals to inhibit the hepatitis virus. It is not practicable for many individuals; however, a liver transplant becomes the only suitable alternative. A liver transplant is an expensive procedure. Thus, there is a significant need to identify potential targets of liver fibrosis and the development of such agents that can effectively treat or reverse liver fibrosis by targeting them. Researchers have identified hypoxia-inducible factors (HIFs) in the last 16 years as important transcription factors driving several facets of liver fibrosis, making them possible therapeutic targets. The latest knowledge on HIFs and their possible role in liver fibrosis, along with the cell-specific activities of such transcription factors that how they play role in liver fibrosis progression, is discussed in this review.

## INTRODUCTION

Liver fibrosis develops when the liver undergoes any damage or injury and is considered one of the major clinical concerns worldwide. Alcohol consumption, drugs, viral infections, genetic disorders, fatty liver diseases like non-alcoholic, alcoholic fatty liver diseases, and autoimmune hepatitis are major contributing factors to liver injury. Hepatocellular injury caused by all these factors starts a mechanism of reparation. This process activates the differentiation of hepatic stellate cells (HSCs) to myofibroblasts. This differentiation process of HSCs is known as "activation" (*Friedman, 2008*; *Sun & Kisseleva, 2015*). Once this happens, proliferation and migration of these cells begin along with collagen synthesis (*Karin et al., 2016*). This collagen acts as a first repair matrix. The excessive collagen eliminates after a severe liver injury, and the myofibroblasts regress to an inactive form. However, if hepatic damage became persistent, collagen begins to accumulate (*Nishio, 2019*; *Dranoff & Wells, 2010*).

Corresponding author
Shizhu Jin, drshizhu-jin@hrbmu.edu.cn

This excessive collagen accumulation leads to fibrosis and inevitably to cirrhosis, which may cause loss of normal functions of the liver or the development of cancer (*Saber, 2018*; *Hernandez-Gea & Friedman, 2011*). Annually, two million human beings die from decompensation and hepatocellular carcinoma (HCC) caused by liver cirrhosis, accounting for over 3.5 percent of all fatalities around the world. HCC is the most generic form of liver-related cancers and 2nd prominent cause of cancer-related deaths. Data from the earlier studies have shown that rate of liver cancer affecting humans increasing day by day but there are only a few treatment strategies developed for early and late-phase fibrosis. The common liver disorders which result in fibrosis are alcoholic liver disease, hepatitis B & C, and NAFLD. Non-alcoholic fatty liver disease (NAFLD) most often progresses into non-alcoholic steatohepatitis (NASH) (*Anstee, 2019*).

According to research by *Holzner & Murray (2021)* HIFs, particularly HIF-2, have a role in promoting steatosis. Studies on the activation of HIF in normoxic settings imply that HIF-2 can impede fatty acid oxidation (FAO), but studies on the effects of oxygen treatment or antagonists on the activation of HIF-2 in NAFLD suggest that HIF-2 promotes lipogenesis. These processes may account for the inhibitory impact of HIF-2 deletion on steatosis in NAFLD. While HIF activation has been linked to certain positive outcomes, such as a potential function in increasing insulin sensitivity, overall, HIF activation is detrimental in NAFLD and may thus be a suitable therapeutic target (*Holzner & Murray, 2021*).

In these cases, fibrinogenesis progresses because of recurrent liver injury. Fibrosis is a restorative mechanism that disrupts the extracellular matrix's (ECM) homeostasis and contributes to an abnormal connective tissue deposition in the liver over time. Immune incursions with impaired angiogenesis are common conditions in liver fibrosis. Collectively these features change the structure of the liver, reduce its elasticity, and gradually cause deterioration of organ structure and function. Moreover, when connective tissue encapsulates the healthy liver tissue it leads to the formation of nodules during late-stage fibrosis (cirrhosis). Consequently, the formation of nodules develops a condition that is characterized by liver stiffness, portal hypertension with abnormal blood flow, and liver decompensation or HCC resulting in higher rates of mortality (*Khomich, Ivanov & Bartosch, 2020*).

Several regulatory agents have been recognized that control expression of HSC (*Moreira, 2007*; *Abbas et al., 2020*), however, the pathways through which regulation of these mediators' production in case of acute and chronic damage is controlled are not well studied. Fortunately, new research has shown that for this process, might be a group of transcription factors (TFs) is needed. These TFs are known as hypoxia-inducible factors (HIFs) (*Moon et al., 2009*; *Copple, Kaska & Wentling, 2012*).

*Ju, Colgan & Eltzschig (2016)* in a review suggested that the protection of cells, stimulation of angiogenesis, and restructuring of cellular energy metabolism were all brought about by the transcriptional actions of HIF. However, the same HIF activities led to a detrimental involvement in the development of tumors, hepatic lipid buildup, and fibrogenesis in chronic liver disorders. These findings suggested that, while HIF suppression helps prevent and treat chronic liver disease, HIF activation was preferable in

cases of acute liver damage. The HIFs' dual function suggested that the therapeutic effects of HIF modifiers could be disease- and time-dependent. For instance, switching from using HIF inhibitors to activating HIFs briefly may be helpful in HCC patients undergoing resection or liver transplantation. Patients with liver cirrhosis who experience an episode of acute-on-chronic liver damage may consider the same treatment approach.

Such oxygen sensing systems have been evolved by organisms that help them to respond to varying oxygen levels. To sustain a homeostatic environment in the body, these sensing systems induce appropriate and efficient transcriptional responses. Complexes of HIF transcription factors are used by these systems. HIFs are highly conserved heterodimers formed of alpha ($\alpha$) and beta ($\beta$) subunits (*Palazon et al., 2014*; *Wang et al., 1995*). Hypoxia can be defined as "decreased cellular oxygen levels". In a hypoxic cellular environment, expression of HIF results in increased regulation of different genes associated with cell activities like genes involved in metabolism, proliferation, and migration process for sustaining cellular homeostasis. Currently, there have been three HIF transcription factors are found and named HIF-1 alpha (HIF-1$\alpha$), HIF-2 alpha (HIF-2 $\alpha$), and HIF-3 alpha (HIF-3 $\alpha$) (*Roth & Copple, 2015*).

In a study by *Zhan et al. (2015)*, it was concluded that as hepatic fibrosis progressed, there was mounting evidence that HIF-1 $\alpha$ may be regulated by intricate signaling networks that controlled its expression. These investigations supplied crucial insights into the effects of HIF-1$\alpha$ over-expression in hepatic fibrosis. Most impressively, new, and profound insights into HIF-1$\alpha$ role in treating hepatic fibrosis were revealed by breakthroughs of its function and mechanism in the development of hepatic fibrosis. It was amazing to find that HIF-1 $\alpha$ 's feedback loop signaling pathway showed several interesting treatment approaches for several disorders, including HIF-1 and VEGF inhibitors. Furthermore, HIF-1$\alpha$ activity was strongly correlated with epigenetics, which suggested potential therapeutic targets. Also, there was an indirect link between TGF-$\beta$, NF-k $\beta$ pathways, and HIF-1$\alpha$ (*Zhan et al., 2015*).

*Chu et al. (2022)* examined that there is growing evidence that HIF-1 may be engaged in intricate signalling networks to control its own expression in a range of liver disease processes. HIF-1 is widely implicated in the occurrence, development, and prognosis of numerous liver disorders (*Chu et al., 2022*).

Several regulatory agents have been recognized that control expression of HSC (*Moreira, 2007*; *Abbas et al., 2020*); however, the pathways through which regulation of these mediators' production in case of acute and chronic damage is controlled are not well studied. Fortunately, new research has shown that for this process, might be a group of transcription factors (TFs) is needed. These TFs are known as hypoxia-inducible factors (HIFs) (*Moon et al., 2009*; *Copple, Kaska & Wentling, 2012*).

Such oxygen sensing systems have been evolved by organisms that help them to respond to varying oxygen levels. To sustain a homeostatic environment in the body, these sensing systems induce appropriate and efficient transcriptional responses. Complexes of HIF transcription factors are used by these systems. HIFs are highly conserved heterodimers formed of alpha ($\alpha$) and beta ($\beta$) subunits (*Palazon et al., 2014*; *Wang et al., 1995*). Hypoxia can be defined as "decreased cellular oxygen levels". In a hypoxic cellular environment,

expression of HIF results in increased regulation of different genes associated with cell activities like genes involved in metabolism, proliferation, and migration process for sustaining cellular homeostasis. Currently, there have been three HIF transcription factors are found and named HIF- 1 alpha (HIF-1α), HIF-2 alpha (HIF-2 α), and HIF-3 alpha (HIF-3 α) (*Roth & Copple, 2015*). Among these three factors, HIF-1α and HIF-2 α have received the most attention. While HIF β also has three paralogues: aryl hydrocarbon receptor nuclear translocator (ARNT) or HIF-1β, HIF-2 β (ARNT 2), and HIF-3 β (ARNT 3). Before controlling the transcriptional activity of genes, subunits of HIF-1α, as well as HIF-2α, heterodimerize with ARNT (*Roth & Copple, 2015*; *Mandl & Depping, 2014*). ARNT is expressed ubiquitously and available in excessive amounts, while the alpha subunit's transcription depends on the oxygen levels.

Normally when oxygen is available sufficiently, the 26S proteasome instantly targets the HIF-1α subunit for degradation purpose (*Kong et al., 2007*). While hypoxic situations lead to inhibition of HIF- α subunits degradation and its stabilization and nuclear translocation. Dimerization with HIF-1β begins soon after HIF-1α enters the nucleus, a HIF complex is formed as a result. In target genes, this complex binds to hypoxia-responsive factors (*Wang et al., 1995*). Thus, HIFs cause hepatic fibrinogenesis when the liver experiences hypoxia in response to liver damage or injury.

Present treatment options for liver fibrosis include temporary relief by the elimination of extra body fluid and the use of certain antibiotics as well as lactulose. Liver transplantation stays the most appropriate option to cure liver fibrosis, but it is quite expensive for many people, it is difficult to find a donor and it has post-transplantation immunosuppressive complications. Other effective and affordable curative strategies are needed for liver fibrosis prevention. As the involvement of HIF-1α in the etiology of hepatic fibrinogenesis is well reported because of its pathological activity, targeting HIF will be a promising choice to impede the development of fibrosis and cancer. This review will shed light on how HIF causes hepatic fibrinogenesis and how it can function as a therapeutic target.

## Hypoxia and hypoxia-inducible factors

Since most organisms need ATP for metabolic processes, oxygen must be present for ATP to be produced in adequate levels (*Rodriguez et al., 2008*). Cells and tissues in the human body experience hypoxia, which is caused by an oxygen shortage. Liver damage, blood circulation troubles, lung conditions, and heart illness are just a few of the causes for this condition. The signaling cascade of hypoxia is triggered by oxygen deprivation and is controlled by the HIF's stability. Based on the degree of the hypoxic situation, permanent damage to bodily tissues may occur (*Semenza, 2000*). Hypoxia can nevertheless serve important and helpful purposes in the physiology and growth and development of humans. The appropriate growth of an embryo depends on it, thus yes, it is needed. Oxidative stress has been linked to the development of the central nervous system, the stimulation of cell death, and the appropriate growth of structures during pregnancy (*Webster & Abela, 2007*; *Dunwoodie, 2009*), even if the precise mechanisms by which it works are unknown. These findings imply that environmental cues, such as hypoxia, function in addition to genetic cues to influence embryogenesis (*Adelman et al., 0000*; *Ziello, Jovin & Huang, 2007*). Many

organisms have evolved adaptive defenses against hypoxic circumstances. Some energy homeostasis regulating genes may be triggered or suppressed as a result of varying oxygen levels, allowing tissues and cells to survive despite changes in their environment. Enzymes and other transcriptional factors may interfere with the control of tissue regeneration and tissue development by genes such as HIF-1, whose regulation is increased by a hypoxic environment (*Ziello, Jovin & Huang, 2007*).

Prolyl-4-hydroxylases (*Hashimoto & Shibasaki, 2015*) manage the hydroxylation of -subunits at proline residues in normoxic conditions. When the HIF subunit connects with Von Hippel-Lindau (VHL) following hydroxylation and promotes ubiquitination, this results in the instability of HIF protein through proteasomal destruction. An E3 ubiquitin ligase is VHL (*Zurlo et al., 2016*; *Mole et al., 2001*). Additionally, factor-inhibiting HIFs aid in the hydroxylation of asparagine residues of the -subunits of HIF (FIHs). Because it inhibits HIF and connects it to CREB-binding protein/p300 (*Schofield & Ratcliffe, 2004a*). It represses HIF activity. The hydroxylation of HIF- subunits is inhibited in conditions of oxygen deprivation or disturbances in the cellular redox balance (hypoxic environment), which results in a buildup of HIF- subunits and diminished PHD and FIH activity. After HIF and accumulated HIF-subunit dimerization, the complex travels in the direction of the nucleus. The transcriptional activity of the heterodimer HIF and HIF increases in the nucleus when it binds to hypoxia-responsive elements (HREs) found on the promoter region of its target gene (*Strowitzki, Cummins & Taylor, 2019*). Vascular endothelial growth factors (VEGF), among many other target genes, are under the control of HIF-1 and HIF-2, along with certain interactions between the two factors (*Kim et al., 2006*). Other methods that have been thoroughly researched have been implicated in the control of post-translational HIF (*Dimova & Kietzmann, 2010*).

## HIFs regulation in an oxygen-dependent manner

As a result of coordinated gene expression following transcription, cells react to decreased oxygen levels. HIFs are part of a group of proteins that make up the PAS (Per-Arnt-Sim) protein family (*Semenza, 2007*; *Goda & Kanai, 2012*; *Wilson, Tennant & McKeating, 2014*). HIFs regulate a variety of signaling processes when they bind to the HREs of the target gene. The target gene's transcription is after up- or down-regulated because of this binding. The HIF group's three transcription factors, HIF-(1-3), are further made up of the heterodimeric subunits and. While the expression of the -subunit is constitutive, the -subunit is activated by proteolytic degradation in the presence of oxygen. HIF-alpha is stabilized in hypoxic environments, while PHD 1-3 destroys this subunit at normal oxygen levels (*Berra et al., 2003*; *Appelhoff et al., 2004*) (Figs. 1A and 1B). Particularly, loss of PHD-2 in the liver reveals that HIF-1 is expressed continuously, while deletion of PHD-3 causes persistent transcription activity of HIF-2, highlighting the variable character of the HIF-regulation mechanism. When E3 ubiquitin ligase (p VHL) finds hydroxypropyl residues and leads to the polyubiquitylation of the HIF alpha subunit, proteasome degradation takes place. Reduced PHD activity in hypoxic settings leads to stability of HIF-1 as well as attachment to its transcription factors (*Schofield & Ratcliffe,*
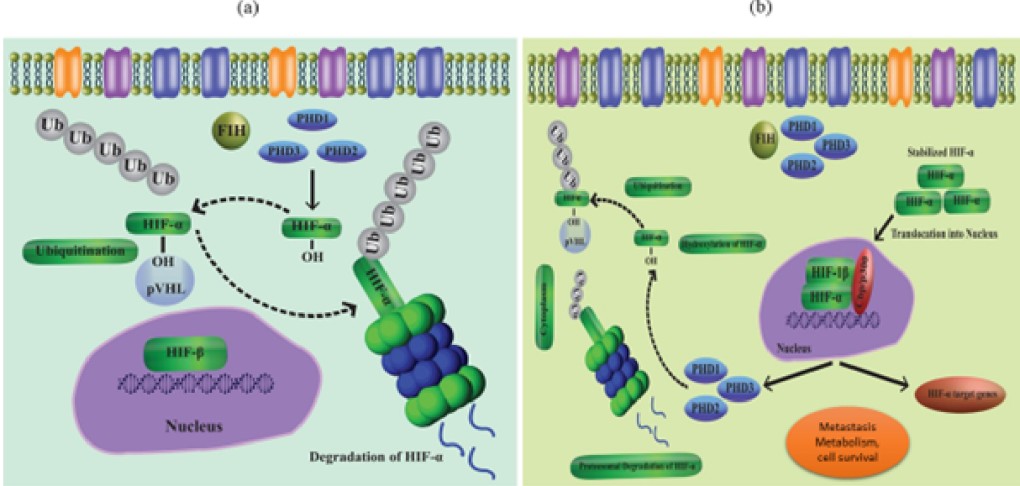

**Figure 1** (A) Regulation of HIF in normal conditions (normoxia), (B) regulation of HIF in hypoxia conditions.

*2004b*) *via* its C and N terminal transactivating sites. CBP and HIFs bind together to create the scaffold of the transcriptional activator.

HIF-1 induced feedback-loop increases the transcription of PHD over the protracted duration of decreased oxygen levels, as described during illness states, leading to recurrence of HIF-1 hydroxylation and depletion (*Ginouves et al., 2008*). While HIF-2 expression can increase in hypoxia over time and support chronic hypoxic conditions (*Patel & Simon, 2008*), HIF-1 activation may be an initial low pO$_2$ response.

The important oxygen-sensing hydroxylase FIH is another one that controls the transcription of HIF-1. In the transcription factor scaffold domain, also known as the transactivating domain or N803, FIH hydroxylates the asparagine residue of HIF-1 found at the C-terminus (in humans). p300, a transcription-coactivating protein that aids in increasing the expression of the target gene, is after inhibited from interacting with HIF-1 dimer as a result (*Sim et al., 2018*; *Lando et al., 2002*). FIH42 is not involved in the hydroxylation of HIF-2. According to reports, PHDs have a low affinity for oxygen, with *in vitro* tests proving detectable Km values of 230–250 M, or well over 21% O$_2$, compared to FIH at 90 M, or around 8% (*Koivunen et al., 2004*; *Hirsilä et al., 2003*).

Since FIH suppresses the activity of HIF-1α's C- terminus transcription factor domain but not the N- terminus domain, it has significant consequences for the hallmark of hypoxia-induced transactivation of genes (*Dayan et al., 2006*). This proves that HIF-1 may control a cell's transcriptional profile in two stages: under moderate hypoxia circumstances, a PHD-inactivation dependent transcriptional profile runs, but during acute hypoxic conditions, both PHD and FIH inactivation dependent transcriptional profiles operate (*Wilson, Tennant & McKeating, 2014*).

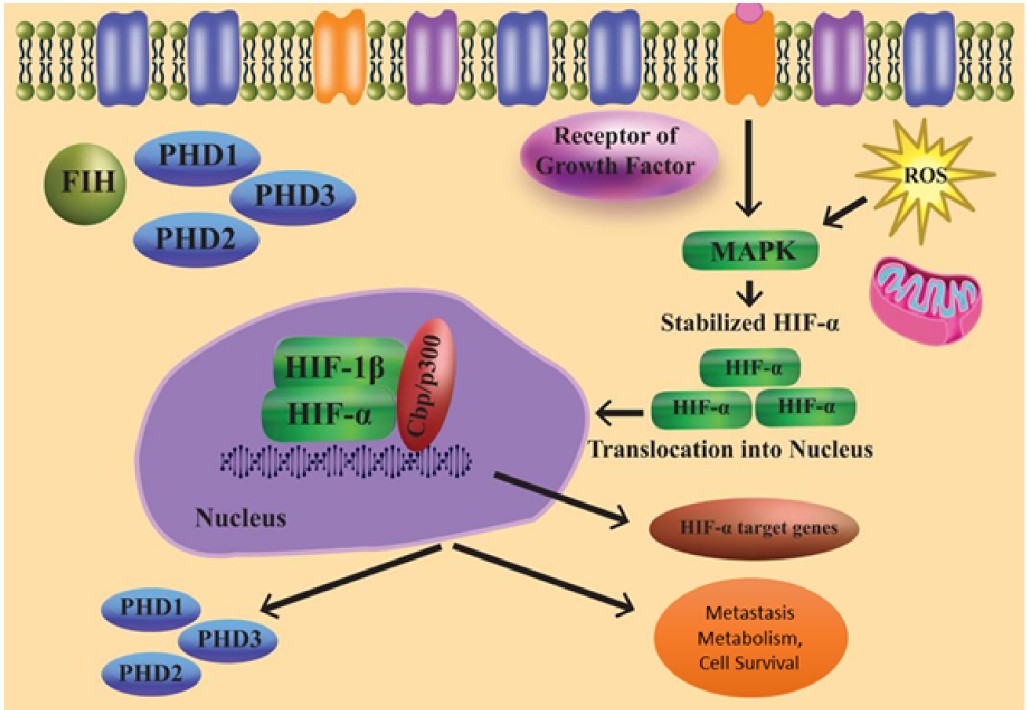

**Figure 2** Regulation of HIF independent of oxygen.

## HIFs regulation in an oxygen-independent manner

As a result of several oxygen-independent signalling events and cellular stress in the presence of oxygen, the state known as "pseudohypoxia" develops (Fig. 2). HIF-1 mRNA translation and transactivation mechanisms are supported by cell surface receptors such GPCRs and RTKs. The HIF regulation system, which is not oxygen-dependent, may be connected to the microenvironment. In the PI3K pathway, phosphatidylinositol-3-kinase (PI3K) is activated when insulin-like growth factor-1 (IGF-1) binds to its cell surface receptor. PI3K controls the downstream part protein kinase B as well as the mechanistic target of rapamycin (mTOR). Because the concentration of local amino acids controls the mTOR activity, this sets up a connection between HIF signalling and the microenvironment (*Wilson, Tennant & McKeating, 2014*).

Furthermore, ribosomal protein S6 kinase 1 and ERK (extracellular-signal-regulated kinase) are necessary for the translation of HIF-1 mRNA. These might be triggered by growth factors. Instead of activating ribosomal protein S6 kinase-1, ERK can stimulate the Mitogen-activated protein kinase (MAPK)-interacting protein through eukaryotic translation initiation factor 4E (eIF4E) activation and mTOR *via* inhibition of the 4E-binding protein (*Wilson, Tennant & McKeating, 2014*; *Semenza, 2003*; *Agani & Jiang, 2013*).

## Expression of HIFs in the liver during hepatic injury

Many reports have described that hepatic hypoxia can develop because of serious liver damage. Development of hypoxia was seen in the liver after alcohol treatment, according to early research (*Roth & Copple, 2015*; *Li et al., 2004*; *Bardag-Gorce et al., 2002*). later studies have shown that compounds like monocrotaline or acetaminophen also cause a hypoxic condition in the liver. The pathway through which this develops is not identified (*Copple, Ganey & Roth, 2003*; *Sparkenbaugh et al., 2011*). However, mostly it occurs due to disturbing the liver structure that results in impeding the flow of blood by disrupted areas; stimulation of the blood coagulation leading to the formation of fibrin clots in the vasculature; and the release of vasoactive mediators modulating the flow of hepatic blood to damaged regions (*Copple, Roth & Ganey, 2006*). Most of these research studies have used a compound named pimonidazole, labeled as a hypoxic probe, to recognize regions of hypoxia (*Roth & Copple, 2015*; *Arteel, Thurman & Raleigh, 1998*). In contrast to this technique, HIF-1 α signaling was being used as a hypoxia substitute marker. Ethanol, acetaminophen, or carbon tetrachloride (CCl4) treated mice depicted that HIF-1α was present in their liver (*Li et al., 2004*; *James et al., 2006*).

However, activation of HIF-1α other mediators like growth factors, oxidative stress, and cytokines in addition to hypoxia are also involved. Therefore, prior to correlating HIF-1 α activation with hypoxia, caution must be used.

Expression of HIF-1α in the liver started when treated with acetaminophen and CCl4 even prior to hypoxia. It showed that when the liver is exposed to such toxic substances other pathways may also regulate the activation of HIF-1α (*James et al., 2006*; *Mochizuki et al., 2014*). Just like acute injury, studies have proved that even after chronic injury, the presence of hypoxia is found in the liver.

Rosmorduc et al. (*Corpechot et al., 2002b*; *Rosmorduc et al., 1999*) first times proved hypoxia development in the liver when they administrated diethyl nitrosamine to rats or even after bile duct ligation. In both these cases, the development of acute fibrosis was seen. These findings were later validated by *Moon et al. (2009)*. Their study showed that after bile duct ligation, activation of HIF-1α is stimulated in many types of cells in the liver (*Moon et al., 2009*). In particular, the expression of HIF-1α was present in immune cells and hepatocytes found inside and on the peripheral sites of necrosis. The presence of hypoxia was seen in both these areas.

In contrast to animal models, they saw the existence of HIF-1 α protein in individuals suffering from primary biliary cirrhosis. HIF-1α was found in hepatic cells as well as scar-associated macrophages close to areas of fibrosis in the livers of these patients (*Copple, Kaska & Wentling, 2012*). HIF-1α was also found to express myofibroblasts in alpha-smooth muscle actin (α-SMA) within sites of fibrosis (*Copple, Kaska & Wentling, 2012*). It has been reported that the α-subunit of HIF-1 activates many genes having a significant role in the progression of fibrosis. Platelet-derived growth factor (PDGF), fibroblast growth factor-2 (PGF-2), VEGF, plasminogen activator inhibitor- 1 (PAI-1), and many others have genes that have been reported (*Yoshida et al., 2006*; *Calvani et al., 2006*; *Stoeltzing et al., 2003*; *Fink et al., 2002*). These studies showed the potential role of HIF-1 α in liver fibrosis progression.

## Profibrotic function of HIF in hepatocytes

As mentioned before, activation of HIF-1α was seen in hepatic cells of mice after they were exposed to bile duct ligation and in persons suffering from primary biliary cholangitis and primary sclerosing cholangitis (*Moon et al., 2009*; *Copple, Kaska & Wentling, 2012*). *Kietzmann, Roth & Jungermann (1999)* in their study also revealed the expression of HIF-1α in hepatocytes during hypoxia.

They described the involvement of HIF-1α in the regulation of PAI-1 transcription in such hepatocytes. The study conducted by *Copple, Kaska & Wentling (2012)* confirmed these results and explained the activation of HIF-2α in hypoxic hepatocytes. Their study showed that the role of HIF-2α has been crucial for the complete regulation of PAI-1 in such hypoxic hepatic cells (*Copple et al., 2009*).

Their study also described that in addition to PAI-1, hypoxia through HIF-1α, as well as HIF-2α, furthermore induced upregulation of VEGF and adrenomedullin- 1 and 2 (*Copple et al., 2009*). Surprisingly, while regulation of PDGF-A and PDGF-B is controlled *via* HIF-1α in certain cells, the expression of PDGF-A and PDGF-B were not increased in hypoxic hepatocytes. An association between HIF and TGF-β signaling pathways was also reported in their study for the first time. In his research, *Copple (2010)* concluded that hypoxic hepatocytes are involved in the HIF-dependent expression of latent TGF-β1.

While it is not recognized through which mechanism this works, they confirmed that in hepatocytes the expression of many matrix metalloproteinases and thrombospondin-1 increased, all of which can regulate the expression of latent transforming growth factor β1 (*Young & Murphy-Ullrich, 2004*; *D'Angelo et al., 2001*; *Maeda et al., 2001*). Collaboratively, the findings of these *in vitro* studies have described that hypoxic conditions in hepatocytes trigger the expression of HIF-1α as well as HIF-2α which ultimately control the PAI-1 and VEGF activation, and also control the expression of latent transforming growth factor β1, which can facilitate the liver fibrosis progression (Fig. 3) Many laboratories have also evaluated the association of hepatocytic-HIFs with the progression of liver fibrosis after these findings by *in vivo* experiments. *Scott et al. (2015)* described that they generated hepatocyte-specific HIF-1β knockout mice by the crossing of HIF-1β or ARNT fluxed mice with Cre recombinase expressing mice (Cre recombinase expression regulated by albumin promoter). Heterodimerization of HIF-1α with the basic family of transcription factors of helix-loop-helix-PER-ARNT- SIM, including HIFs, occurs as described previously, and HIF-1*β* is, therefore, necessary for transcription activity of both HIF-1α and HIF-2α (*Scott et al., 2015*).

Treatment of hepatotoxicant thioacetamide was given to HIF-1β knockout mice and control mice (without HIF-1β knockout) in hepatocytes to elicit liver fibrosis. In both types of mice, though the pattern of fibrosis was alike, it was seen that macrophage infiltration decreased in hepatocyte-specific HIF-1β knockout mice. Also, the transcriptional activity of profibrotic genes, namely TGF-β1 and TGF-β2, collagen type 1 and 5, and tissue inhibitor metalloproteinases 1 and 5 was decreased in hepatocyte-specific HIF-1β knockout mice (*Scott et al., 2015*). Such findings proved the significance of HIFs in controlling the expression of profibrotic gene expression in hepatic cells *in vivo* after liver fibrosis. Later

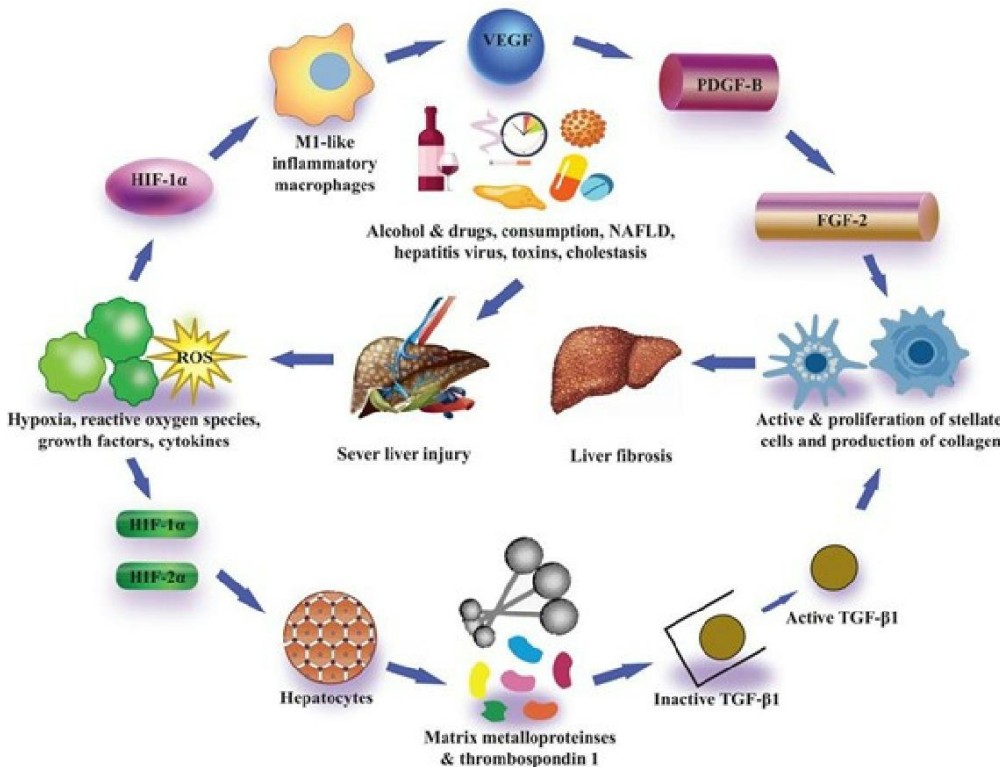

**Figure 3** **Pro-fibrotic function of HIFs in liver fibrosis.**

research by *Roychowdhury et al. (2014)* also indicated the significance of hepatocytic-HIF-1α in fibrosis development. Ethanol along with CCL$_4$ treatment was subjected to hepatocyte-specific HIF-1α-deficient in their study, which resulted in increased liver fibrosis induced by CCL$_4$ (*Roychowdhury et al., 2014*).

As compared to control mice, the progression of liver fibrosis was observed to be decreased in treated mice due to deficiency of HIF-1α in the liver which controlled and inhibited the ethanol and CCL$_4$ promoted liver fibrosis, as revealed by mRNA profile and protein levels of both type-I collagen and α-SMA. Conversely, in hepatocyte-specific HIF-1α deficient mice when only CCL$_4$ treatment was given, liver fibrosis did not reduce. This may reveal that hypoxia generated by the metabolism of ethanol may increase the expression of HIF-1α in hepatic cells after subjecting to CCL$_4$, prompting HIF-1α to play a significant role in liver fibrosis progression (*Roychowdhury et al., 2014*).

*Qu et al. (2011)* reported in their studies that the presence of HIF-2α in hepatocytes may also necessarily be required for fibrosis development. In their studies, they developed hepatocyte specific VHL knockout mice.

Deletion of the VHL gene causes the constitutive expression of HIF-1α and HIF-2α in hepatocytes (*Haase et al., 2001*). Liver mRNA profiles of control and VHL deficient mice were then saw. It has been found that the expression of many genes that are critical for liver fibrosis progression was upregulated. These upregulated genes are essential for collagen

synthesis, like procollagen-lysine 2, oxoglutarate 5-dioxygenase 2, and transglutaminase-2 as well as α-SMA (*Qu et al., 2011*).

Besides, an increase in fibrosis was induced by feeding an ethanol diet to VHL deficient mice. In the mice deficient in VHL as well as HIF-2α (double knockout), the rate of fibrosis was not increased in hepatocytes. However, in VHL-HIF-1α double knockout mice, hepatocytes showed an increase in fibrosis (*Qu et al., 2011*). Collectively, these results showed that activation of HIF-2 α in hepatocytes is critical in this mice model for liver fibrosis progression (*Qu et al., 2011*). Furthermore, this study has one limitation *i.e.,* Although HIF-2 knockout prevented the rapid onset of fibrosis, it also resulted in overall less liver damage. Since impairment of the liver is a key part that causes liver fibrosis, HIF-2α may not directly cause hepatic fibrinogenesis, but instead prompts liver damage. Supplementary research is needed to describe the link of hepatocyte-specific HIF-2α more explicitly and systematically in liver fibrosis development.

## HIF's role in liver fibrosis development

During embryogenesis, the knockout of HIF-1α causes the death of mice. Therefore, to evaluate the idea that HIF-1 α leads to the progression of liver fibrosis, researchers used the Cre-lox technique in adult mice for the knockout of HIF-1 α. In their study, they crossed the HIF-1α fluxed mice with the Cre recombinase-expressing mice. Cre recombinase expression was regulated by interferon inducible Mx promoter (*Moon et al., 2009*). When these mice were treated with polyinosinic-polycytidylic acid it was seen that in most types of cells the expression of Cre recombinase increases ubiquitously (*Roth & Copple, 2015*; *Kuhn et al., 1995*).

Treated mice and control mice that were not subjected to polyinosinic-polycytidylic acid treatment, both were exposed to BDL. Results proved that liver fibrosis was reduced greatly in HIF-1α knockout mice. This showed the potential role and involvement of HIF-1α in liver fibrosis development *in vivo* for the very first time (*Moon et al., 2009*). Knocking out HIF- 1 α inhibited the up-regulation of many primary profibrotic mediators such as PDGF-A and B, PAI-1, and primary fibroblast growth factor-2 (*Koivunen et al., 2004*). All these gene products have been associated with the development of liver fibrosis, showing that by controlling the expression of these genes, HIF-1α can promote fibrosis (*Neef et al., 2006*; *Bergheim et al., 2006*; *Yu et al., 2003*). Besides, reports have proved that these genes in certain cell types are specifically regulated by HIF-1α (*Yoshida et al., 2006*; *Calvani et al., 2006*).

*Mesarwi et al. (2016)* in their study demonstrated the role of HIF-1 in the development of liver fibrosis by evaluating the effect of HIF- 1α deletion on reduced liver fibrosis. The findings of their study suggested that knocking out HIF-1 in hepatocytes markedly reduced liver fibrosis in the NAFLD mice model. These research studies suggested a significant role of HIF-1α in the progression of liver fibrosis, its cell-specific role in the etiology of the fibrotic process of the liver still needs further investigation.

*Han et al. (2019)* proposed that hepatocytes with HIF-1 inactivation may be protected against liver fibrosis. First, they discovered that HFD caused 80% greater hepatic collagen deposition than HIF-1α /hep animals, which was supported by liver tissue stained with

SMA. The weights of the groups' livers and bodies were comparable. Then, they discovered that in mice on the HFD and in HepG2 cells treated with PA, HIF-1α expression was rising while PTEN expression was falling. In the end, they discovered that the HIF-1-mediated PTEN/NF-kB-p65 pathway is crucial for the progression of NAFLD to liver fibrosis. These findings collectively point to a new HIF-1α /PTEN/NF-Bp65 signaling pathway in NAFLD that may be therapeutically addressed (*Han et al., 2019*).

*Tian et al. (2018)* postulated that the HIF signaling pathway might be used to control the expression of CD248/endosialin in HSC, which would after have an impact on the function of HSC and the onset of liver fibrosis. Two significant studies were conducted to test the validity of the hypothesis: (1) in the BCS animal model and clinical studies, the relationship between liver fibrosis severity and the expression of HIF and CD248/endosialin in HSC was investigated; and (2) in the *in vitro* cell system, the influence of the hypoxic microenvironment, HIF-1 or HIF-2, on the expression of CD248/endosialin in HSC was investigated. It was discovered that the HIF signaling pathway controlled the expression of CD248/endosialin, causing the onset of liver fibrosis caused by BCS (*Tian et al., 2018*).

*Zou et al. (2017)* discovered that there should be variation in how HIFs affect the risk, clinicopathological characteristics, and survival of different human malignancies. The molecular heterogeneity of cancer and the impact of hypoxia on various kinds of human tumors might be considered plausible explanations. Added research should reveal the potential mechanism (*Zou et al., 2017*).

Recent research by *Foglia et al. (2021)* has shown that there are two main HIF subunits, HIF-1, and HIF-2 (and then HIF-1 and HIF-2), may both respond to hypoxia, but that their transcriptional gene targets, oxygen dependency, and activation kinetics can differ significantly. A paradigm where HIF-1 and HIF-2 may function differently in connection to chronic liver disease (CLD) development is beginning to emerge from preclinical and clinical investigations, with a current emphasis mostly on progressing NAFLD and alcoholic liver disease (ALD) (*Foglia et al., 2021*).

*Yuen & Wong (2020)* concluded that HIFs in Kupffer cells in the early phases of hepatocarcinogenesis enhance liver cirrhosis by activating HSCs to produce profibrogenic substances. HIFs function as master regulators in the later stages of hepatocarcinogenesis to suppress the liver's innate immunity, enabling cancer cells to escape immune detection. Together, innate immunity and adaptive immunity influence how the body reacts to immunological therapy. By transcriptionally activating genes crucial for Treg enrichment and T cell fatigue, HIFs also direct the adaptive immune system (*Yuen & Wong, 2020*).

## Hypoxic Kupffer cells and liver fibrosis

The Kupffer cells are native liver macrophages. Even though these cells exhibit a key role, they are key mediators to promote a broad range of hepatic disorders, like ischemia-reperfusion damage, liver injury caused by toxins, fatty liver disease, and liver fibrosis (*Rivera et al., 2001*; *Jaeschke & Farhood, 1991*). Kupffer cells or existing hepatocellular macrophages are crucial for controlling inflammation through the hepatic fibrosis process. Reactive oxygen species (ROS), and several pro-inflammatory cytokines, including tumor necrosis factor- α, interleukin- 1 β, and macrophage inflammatory protein- 1, are secreted

by Kupffer cells. Such a cell could prompt hepatic stellate cell (HSC) expression. HSC then generates TGF-β and PDGF pro-fibrotic cytokines and leads to liver injury (*Liu et al., 2010*; *Pradere et al., 2013*). By chronic treatment of CCL$_4$ in rats, *Rivera et al. (2001)* first demonstrated the crucial role of macrophages in liver fibrosis progression. These findings have been supported by later research studies, confirming the implication of macrophages in hepatic fibrosis (*Duffield et al., 2005*).

The findings of *Karlmark et al. (2009)* research study described that the recruitment of the macrophage's population, which induced fibrosis, from the bone marrow occurred in a C-C chemokine type 2 and type 6 dependent manner. These cells' immunophenotyping revealed that they were macrophages originated from CD11b +F4/80+Gr1+ monocytes. Even though these results showed the critical implication of macrophages in hepatic fibrosis, how these macrophages start and provoke the fibrosis was unclear. Preliminary work by *Friedman & Arthur (1989)* revealed that in a manner based on PDGF, a controlled medium from cultured Kupffer cells promoted proliferation of HSC in culture.

Furthermore, studies have shown that synthesizing collagen CD11b+F4/80+Gr1+ macrophages present inside the hepatocytes can activate HSCs in a TGF-β-based manner (*Karlmark et al., 2009*). Following these findings, histochemical studies and *in situ* hybridization approaches have shown that the expression of TGF-β and PDGFs is regulated by macrophages in the liver during fibrogenesis (*Nakatsukasa et al., 1990*; *Faiz Kabir Uddin Ahmed et al., 2000*). These studies have described the fact that in the development of fibrosis, macrophages are essential and by expressing growth factors they promote HSC proliferation that can lead to fibrosis. But the mechanism is unclear through which liver damage induced such cells to release profibrotic mediators.

In hepatic fibrinogenesis, kupffer cells also regulate the expression of several chemokines and matrix metalloproteinases that employ immune cells and help the breakdown of the ECM, thereby promoting the recovery of fibrosis (*Baeck et al., 2012*). The abundance of circulating Ly6Chi inflammatory monocytes within the liver is strongly reliant on the CCR2/CCL2 and CCL1/CCR8 axis (*Baeck et al., 2012*). Then, activation of rapidly dividing hepatocytes and MAP3K14 contribute to the production of various chemokines inside hepatocytes. The recruitment/initiation phase of macrophages that so cause cell death of hepatic cells may be affected by these chemokines. Likewise, severe liver inflammation and hepatic cells' death-causing hepatic fibrosis are stimulated by MAP3K14 *in vivo*. The possibility that the reduction of Kupffer cells using clodronate reverted MAP3K14-induced injury according to the above observations (*Irvine et al., 2014*; *Shen et al., 2014*).

Some observations suggested that activation of CX3CR1 on Kupffer cells contributed to increased interleukin 10 (IL-10) transcriptional activity and decreases their TNF and TGF-β (*Aoyama et al., 2010*), whereas IL-10 is considered to show anti-inflammatory appropriateties. It has been found that IL-10 suppresses TNF-α and IL-1 expression and inhibits NF-$\kappa$B expression. IL- 10 decreases the release of macrophages induced by nitric oxide and ROS intermediates and decreases activation of cell surface binding proteins and chemokines (*Gao, 2012*). Therefore, the CX3CR1 ligand reflects negative feedback on the dissemination of inflammation of the liver by affecting Kupffer cells.
As mentioned previously, studies explained that both mice and humans suffering from liver fibrosis showed the activation of HIF-1α in scar-associated macrophages (*Moon et al., 2009*; *Copple, Kaska & Wentling, 2012*). In later studies, they presented HIF-1α activation *via in vitro* exposure of Kupffer cells to hypoxia, but no activation of HIF-2α did not see. These results showed that HIF-1α signaling may prevail in Kupffer cells from healthy livers (*Roth & Copple, 2015*; *Copple et al., 2009*). Nevertheless, in infected livers, this may change. Macrophages differentiate into various phenotypes in the presence of some cytokines, such as M1 (classically activated macrophages) and M2 (alternatively activated macrophages).

A research study by *Takeda et al. (2010)* described the up-regulation of HIF-1α activation in M1 while HIF-2α expression was down-regulated in M2. In healthy, non-infected livers, kupffer cells are shifted towards an M1 phenotype. It may be the reason behind HIF-2α's down-regulated expression in such cells. It has been suggested that, in infected livers, activation of HIF-1α or -2 α transcriptional factors in hepatic macrophages may be due to the injury *in vivo* (*Roth & Copple, 2015*). As HIF- 1α and -2 α both show distinct roles to regulate several dual-coding as well as discrete groups of genes, it would be remarkable to analyze the expression pattern of genes in hypoxic Kupffer cells either shifted toward an M1 phenotype with activated HIF-1α or an M2 phenotype with activated HIF-2α. The possibility of M1 and M2 macrophage populations playing distinct parts in hepatic disorders is present due to varying regulation of TFs (HIF- 1 α and HIF-2α).

Research findings have proven that profibrotic mediators are generated *in vitro* by hypoxic kupffer cells. One study revealed that when Kupffer cells were exposed to a hypoxic environment increased activation of many genes such as VEGF angiopoietin- 1 in a HIF-dependent manner was seen. These genes engage in the angiogenesis process (*Sparkenbaugh et al., 2011*). Furthermore, up-regulation of PDGF-B was also seen in a HIF-dependent manner in hypoxic Kupffer cells (*Copple et al., 2009*). In macrophages, selective knockout of HIF-1β through Cre/lox did not influence the hepatic damage, markers of cholestasis, or hepatocellular inflammation with increased activation of cytokines, hepatic neutrophils, and proliferation of macrophages in mice subjected to bile duct ligation (*Kong et al., 2007*). However, knocking out of HIF- 1 β reduced liver fibrosis as estimated by levels of mRNA and protein of type I collagen and α- SMA (*Copple, Roth & Ganey, 2006*; *Pradere et al., 2013*).

Hence, higher PDGF-B mRNA and protein levels were inhibited by the deletion of HIF-1β in macrophages (*Copple, Kaska & Wentling, 2012*). It also inhibited increased activation of FGF-2, which is one more growth factor involved in pro-fibrinogenesis, however, protein levels of actively expressing TGF-β were not affected by it *Kong et al. (2007)*. Mixed outcomes were reported in mice where the HIF-1α gene was cut in macrophages, showing that HIF-1α is profibrotic in macrophages (*Copple, Kaska & Wentling, 2012*). This is reflective of the studies showing that M1 inflammatory macrophages induce fibrosis (*Karlmark et al., 2009*). In M1 macrophages, the expression of HIF-1α is high, as previously stated. However, it would be significant to study the involvement and function of HIF-2α in macrophages and ascertain whether it has a countering effect and eases the remission of fibrosis.

## Hepatic stellate cells (HSCs) and HIF

HSCs, a population of mesenchymal cells that make up 5–10% of the actual count of the liver cells, are the type of cells significantly involved in hepatic fibrinogenesis. Although the term was firstly reported in the 19th century, its role was proved in 1985 by *Friedman & Arthur (1989)*. Who found that HSCs were the liver's primary type of collagen-producing cells? HSCs are present in the perisinusoidal region where neighboring hepatocytes and endothelial sinusoidal cells surrounds them *Friedman (2008)* and *Takeda et al. (2010)*. The key function of HSCs is to secrete some essential secretions such as the secretion of important ECM proteins.

These ECM proteins are structurally important protein laminin, glycosylated mucoproteins, and collagen IV (Col4) which help to form the backbone of basement membrane-like structures. Proliferation and trans differentiation of inactive HSCs into contractile myofibroblasts start due to paracrine stimulation *via* adjacent cells, such as Kupffer cells, hepatocytes, platelets, leukocytes, and sinusoidal endothelial cells. HSCs become activated and their proliferation is stimulated by Kupffer cells under the influence of various cytokines. Transforming growth factor β 1, lipid peroxides, IL- 1, TNF, and ROS are particular cytokines involved in the stimulation of HSCs (*Friedman, 2008*; *Marrone, Shah & Gracia-Sancho, 2016*). In liver inflammation, the main factor involved in inflammation is lipid peroxides. Platelets release pro-fibrotic growth factors including PDGF, transforming growth factor β 1, and epidermal growth factor. Neutrophils are also considered an essential constituent of ROS (*Semenza, 2007*; *Marrone, Shah & Gracia-Sancho, 2016*). Another source of pro-fibrotic cytokines is lymphocytes (*Khomich, Ivanov & Bartosch, 2020*).

Current scientific investigations using advanced cell-tracing methods have further defined the role of such cells in the deposition of collagen amid the fibrosis process. In an inactive state, accumulation and release of retinoids occur by these cells. After a liver injury, activation, and differentiation of HSCs into such myofibroblasts that express α-SMA have been observed after liver damage. These myofibroblasts then produce collagen, multiply, and circulate all through the liver (*Sun & Kisseleva, 2015*).

HIF-1α is a critical pro-fibro genic agent and it is activated by peripheral hypoxia, oxidative stress, or toxicity. It regulates the transcription of pro-fibrotic factors and enzymes including PDGF, FGF, and PAI- 1, and stimulates the activation of lectin-type oxidized LDL receptor 1 (Lox1) and metalloprotease, which subsequently control the expression of latent TGF-β-1 (*Roth & Copple, 2015*).

Additionally, it is critical for the regulation of metabolic reprogramming, as proven by the up-regulation of the glucose transporter Glut 1, hexokinase 2, pyruvate kinase, and pyruvate dehydrogenase kinase 3 expression, HIF-1α stimulates the glycolytic process. Subsequently, it stops the pyruvate to enter the Krebs cycle and therefore promotes the pyruvate to lactate conversion (*Puche, Saiman & Friedman, 2011*).

HSCs-mediated lactate synthesis in an extracellular environment or produced by transformed hepatocytes results in the accumulation of lactate (*Chen et al., 2012*). This accumulation has the potential to metabolically reprogrammed and activate the adjacent

hepatic stellate cells and may alter the immune environment, as has been shown in the case of other types of cancer (*Chan et al., 2012*).

Moreover, the production and hydroxylation of collagen also depend on the accumulation of lactate (*Khomich, Ivanov & Bartosch, 2020*). In HSCs higher level of the GLUT4, which is a glucose transporter has also been found to promote glucose uptake when observed in a NASH animal model (*Pavlides et al., 2009*). It was found that the AKT phosphorylation by CYP2E1/leptin/purinergic receptor-X7 has been involved in this mechanism. Increased glucose concentration in HSC culture media has been found to stimulate activation due to the p38-MAPK signaling and ROS generation (*Chandrashekaran et al., 2016*; *Wu et al., 2016*). Glycolysis and lactate accumulation is crucial for trans-differentiation of HSC as showed by several research findings involving vismodegib inhibitor (Hedgehog inhibitor) and curcumin (a glycolytic blocker) (*Puche, Saiman & Friedman, 2011*; *Sugimoto et al., 2005*; *Lian et al., 2016*). Any damage to the liver can result in small portions of hypoxia, therefore Corpechot et al. (*Corpechot et al., 2002b*) analyzed the impact of hypoxia on the collagen production ability of HSCs. It was seen that under *in vitro* conditions the type I collagen is upregulated because of the exposure of rat HSCs to hypoxia. This observation shows that hypoxia can induce collagen formation by HSCs. Additionally, comparative outcomes were likewise seen on account of primary human HSCs (*Copple, Kaska & Wentling, 2012*).

Even though this process needs to be further investigated, current evidence suggests that it occurs to produce a primary structure matrix to help repair mechanisms, like angiogenesis. However, in the case of a chronic liver injury, long-term hypoxia can trigger HSCs to keep producing collagen continually, eventually bringing about fibrosis.

Apart from this, in alternative research, *Shi et al. (2007)* discovered that hypoxia can also up-regulate the production of collagen by LX-2 cell, a human HSC line. It was also saw that hypoxia increases the production of autocrine TGF-β, which in turn up-regulates collagen formation (*Lian et al., 2015*). The same study also showed the upregulation of matrix metalloproteinase-2 and its implication in the TGF-β expression (*Lian et al., 2015*). However, this study did not confirm if HIFs are needed in this process, and if the same process occurs *in vivo*.

In this regard, *Copple et al. (2011)* conducted some experiments to figure out if HIFs are also in action or not in HSCs under hypoxia. The outcomes of those experiments showed that when the primary HSCs of mice were subjected to hypoxic conditions, it resulted in activating both HIF- 1 α and HIF-2α expression. It was likewise seen that the level of expression of many fibrosis development genes is increased because of hypoxia. For example, the expression of prolyl-4-hydroxylase- α1 and prolyl-4-hydroxylase- α2 was increased due to hypoxia, as these enzymes are quite crucial in collagen metabolism because they help in the formation of stabilized collagen triple helices (*Copple et al., 2011*). In the same context, hypoxia also resulted in the upregulation of various genes needed in angiogenesis. Some of those genes manage proteins like placental growth factor, VEGF, macrophage-migration inhibitory factor, and angiopoietin-like-4 (*Copple et al., 2011*).

Also, the hypothesis of angiogenesis being the root cause of the hepatic fibrinogenesis process was intended (*Rosmorduc et al., 1999*); however, it is still a controversial subject

matter. For example, it is seen that VEGF can induce collagen production, proliferation, and migration of HSCs under *in vitro* conditions (*Copple et al., 2011*). In addition to this, it was seen that if VEGF is neutralized, it can result in the reduction of fibrosis in mice, which had CCL$_4$ as an earlier treatment (*Novo et al., 2007*). In comparison to this, recent evidence shows that VEGF is crucial for reversing fibrosis (*Yoshiji et al., 2003*; *Kantari-Mimoun et al., 2015*). Anyways, this antifibrotic capability of VEGF depends on its effect on the sinusoidal endothelial cells however not upon HSCs. Therefore, the significance of VEGF and angiogenesis in liver fibrosis is quite a complex subject matter and needs more focus.

Hypoxia not only increases the proangiogenic mediators but also increases the expression of HSCs activation and migration receptors. For example, hypoxia causes an expression enhancement of the chemokine receptors like Ccr1 and Ccr5 (*Shi et al., 2007*). These receptors are activated by the effect of chemokines and leading to the migration of HSCs. The role of these receptors in fibrosis development was verified when mice with knocked out Ccr1 and Ccr5 receptors displayed decreased fibrosis in the liver (*Yang et al., 2014*). Apart from this, adrenergic receptor α2b and interleukin- 13 receptors α1 are two more receptors that were increased in HSCs under hypoxia. Interleukin- 13 and catecholamines are the chemokines that stimulate these receptors (*Lian et al., 2015*). These receptors are found to stimulate collagen formation in HSCs *in vitro*, however, both routes are crucial for fibrinogenesis progression *in vivo* (*Seki et al., 2009*; *Weng et al., 2009*; *Oben et al., 2003*).

Many genes were up-regulated under the influence of HIF-1α. However, some genes were not influenced by HIF-1α. This suggests that there is also a key role of HIF-2α, or some other regulatory transcription factors influenced by hypoxia to stimulate these genes (*Shi et al., 2007*). It is also quite interesting that hypoxia can stimulate upregulated expression of many genes, particularly in culture-activated HSCs, however, it does not upregulate the expression in quiescent HSCs. This fact proposes that the HSC activation can cause a sensitivity enhancement of those already activated HSCs towards HIF-regulated gene expressions (*Shi et al., 2007*). The reason and the mechanism of these happenings are still unidentified, but it may happen because of epigenetic modifications such as histone modifications or DNA methylation which results in the increased access of HIFs towards such gene promoters that are present in activated HSCs.

Additionally, hypoxia not only increases the expression of the genes described earlier, but also those genes that can affect the overall function of HSCs. Two significant examples of those genes are the Jumonji domain-containing proteins Jmjd1a and Jmjd6 (*Shi et al., 2007*). Jmjd1a handles histone demethylation, and Jmjd6 is responsible for mRNA splicing regulation (*Fallon et al., 2000*; *Webby et al., 2009*). These genes can play crucial roles. Firstly, the up regulation of Jmjd1a could enhance the expression of several genes as it can modify their histones, secondly, the upregulation of Jmjd6 can stimulate alternative splicing and produce proteins with distinct functions that can ultimately increase the chances of fibrosis.

Moreover, hypoxia can also increase the expressions of the regulators of G-protein signaling, such as Rgs2 and Rgds4 (*Shi et al., 2007*). Both the proteins act like GTPase-activating proteins for the α subunits of G-protein, which can decrease the signaling by Gq α, Gi α, and Go α subtypes (*Yamane et al., 2006*). Moreover, the upregulated expression of Rgs affects the signaling of the HSCs' GPCR.

However, the prominence of such a phenomenon in hepatic fibrinogenesis is still unknown. Normally hypoxic HSCs cause an increase in gene expression however an unexpected decrease in the genes and their expressions also occurs in hypoxia. α1-integrin and hepatocyte growth factor (HGF) were the two main genres of interest that were decreased due to hypoxia (*Shi et al., 2007*). HGF has a known prime part in liver regeneration, and various research have revealed that during the regeneration process, the major source of HGF is the HSCs (*Skrtic et al., 1997*; *Skrtic et al., 1999*; *Nejak-Bowen et al., 2013*). In another research in 2002, *Corpechot et al. (2002a)* proved that the reason for HGF reduction in hypoxic HSCs is hypoxia. This study proven that in cirrhotic livers the main reason for the failure of liver regeneration is the reduced expression level of HGF in hypoxic HSCs (*Nejak-Bowen et al., 2013*). α1-integrin is a known collagen receptor; however, various research has proven that in mice with knocked out α1-integrin gene the fibroblasts showed enhanced collagen production (*Corpechot et al., 2002a*). The reason and significance of the reduced α1-integrin level in hypoxic HSCs are not known to date. In addition, the fact whether HIF-1α plays any vital part in α1-integrin and HGF's reduction process or not is also undiscovered.

For checking the impact of the activated HIF and hypoxia upon HSCs functional activity the *in vivo* function of HIF-1α upon HSCs in the liver fibrosis progression was evaluated by numerous studies. In all such research studies, a crossing was done between the HIF-1α fluxed mice and Cre recombinase-expressing mice. Glial fibrillary acidic protein promoter was used for the crossing recombination and in the liver, only HSCs were known to express this promoter. As a result of that crossing, the HIF-1α was cut or knocked out in only HSCs. The set of control mice was given the treatment of CCL$_4$'s single hepatotoxic dosage. This treatment of CCL$_4$ enhanced the expression rate of collagen type I, III, and IV over 48 h of treatment. In the same context, at that time point, because of the deleted HIF-1α in the HSCs, there was not any overexpression of any of the three collagen types in the liver. However, after 72 h of the CCL$_4$ treatment, the livers of the mice which had knocked out HIF-1α in HSCs unexpectedly showed elevated expression of mRNA and protein of type I collagen. Furthermore, the elevated levels of α-SMA 's mRNA and protein in the liver showed an upregulation in the HSC activation (*Mochizuki et al., 2014*).

These observed results were different from what has been seen *in vitro* studies. The results propose that HIF-1α holds a potential role in limiting the HSCs' activation and the collagen's production in *vivo* conditions after a liver injury. Further analysis has showed that macrophage activation in the livers of these mice was unsuccessful, and due to these necrotized hepatocytes remained present in the liver (*Mochizuki et al., 2014*). There is a strong possibility that if those necrotized hepatocytes are not removed from the liver, they can keep HSCs active and thereby increasing the collagen production in mice that have HIF-1α deficient HSCs.

Interestingly, plasminogen and urokinase-plasminogen activator knockout mice also show this phenotype, where the body does not remove necrotic hepatocytes and further adds to the increase of collagen production (*Gardner et al., 1999*; *Bezerra et al., 1999*). Furthermore, HIF-1α is not only crucial for the up-regulation of collagen following an injury, but also for the stimulation of HSCs to start producing HIF-1 α-dependent factor(s)

which can activate macrophages, as they are further necessary to remove necrotized hepatocytes *via* phagocytosis. However, the exact process of this event is unknown, but the increase in expression of HIF-1α in HSCs may regulate the formation of post-injury plasmin.

### The perspective of HIF as a therapeutic target for liver fibrosis

Over the last 16 years, researchers have tried to supply an answer about the exact role of HIFs in fibrosis formation in the liver. Moreover, this study strived to explain the part of HIF-1 α in the control of several profibrotic mediators in HSCs, Kupffer cells, and hepatocytes. However, the functioning of HIF-1α in fibrosis, when expressed in other cells is still unassessed. For example, an important target of HIF, VEGF is up-regulated in cholangiocytes posterior to BDL (*Currier et al., 2003*), but it is still unclear if it is stimulated by HIF-1α. Also, HIF-1α is known to regulate numerous cells of the immune system, namely NK-cells, B-cell, and T-cells, which could be quite important in fibrosis (*Gaudio et al., 2006*; *Zhang et al., 2016*; *Albanese et al., 2020*).

*Albanese et al. (2020)* gave an outline of the crucial stages post-translational alterations play in controlling the stability and activity of HIF. Given that hypoxia is a key microenvironmental part in many facets of life and has a considerable influence in many pathologic aspects. This is reduced to 20 times that of any HIF protein by specifically searching for missense mutations (resulting in amino acid alterations). Pathological alterations in growth factor synthesis or driver mutations, such as those in receptor tyrosine kinase pathways, may promote modification of HIFs that may influence the survival or demise of cancer cells (*Albanese et al., 2020*).

Apart from this, the contribution of HIF-2α in fibrosis is still unclear, as also the type of cells it turns on and expresses. However, it is quite clear that the activation of HIF-1α and HIF-2α has a clear link with cancer development. Because it is reported that cirrhosis can usually progress into hepatocellular carcinomas, therefore it will be quite interesting to examine if HIFs are somehow crucial for this shift of cirrhosis into cancer. Different researchers showed that the activation of HIF- 1 α and HIF-2α in liver cells can cause liver hemangiomas, which can be useful for HIFs to induce cancer in the liver (*Qu et al., 2015*).

Anyways, several gaps need to be filled *via* research in the future. Also, the importance of HIFs' inhibition as a therapy against liver fibrosis is yet to be decided. However, there has been only one study in this regard so far. An inhibitor of HIF-1 α called LW6 was administered two weeks after the liver disease initiation in rat models treated with $CCL_4$ and hepatic artery ligation. The results showed that LW6 reduced the progression of fibrosis in the livers of model rats (*Zhan et al., 2015*). However, before finally about HIFs as potential therapeutic targets of liver fibrosis, the results should be confirmed with different model organisms and with different inhibitors of HIFs.

## CONCLUSION

Liver fibrosis is a major hazard to human health that results from severe liver damage brought on by extrinsic causes like alcohol and drug use as well as intrinsic factors like viral hepatitis and genetic differences. Hepatic fibrinogenesis and the expression of the HIF

transcription factors have been linked. Clarifying the fundamental molecular processes of the HIF has gained importance because of the HIF's expanding dynamics in a number of liver-related issues. Experimental animal models can be used to explore the signalling cascades and gene expression targets that HIFs employ to trigger their functions. Other kinds of liver cells than hepatocytes can contribute to the development of fibrosis. It is crucial to investigate whether cell-specific HIFs have protective or dysfunctional effects in various liver disorders.

This review focuses on how HIF affects potential liver cell types, causing liver fibrosis. Researchers would receive help from the knowledge gleaned from earlier work to better understand the dynamic role of HIFs in liver fibrosis and develop therapeutic strategies to control HIFs. For the treatment of ischemia and cancer, HIF controllers are now being developed and evaluated in clinical studies. Although several of these compounds have lately been the subject of preclinical study for HCC and liver transplantation, translational research must be given top priority in the future years.

### Funding
The authors received no funding for this work.

### Competing Interests
The authors declare there are no competing interests.

### Author Contributions
- Jan Mohammad Omar conceived and designed the experiments, analyzed the data, prepared figures and/or tables, authored or reviewed drafts of the article, and approved the final draft.
- Yang Hai conceived and designed the experiments, analyzed the data, prepared figures and/or tables, authored or reviewed drafts of the article, and approved the final draft.
- Shizhu Jin conceived and designed the experiments, analyzed the data, prepared figures and/or tables, authored or reviewed drafts of the article, and approved the final draft.

### Data Availability
    No raw data was used for this review article.

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
