# Peer review of "Hypoxia-induced factor and its role in liver fibrosis"

_PeerJ, doi:10.7717/peerj.14299_

## Round 0.1 · original submission · Major Revisions

Three reviewers have taken a detailed and in-depth look at the manuscript and based on the comments, I advise you to revise thoroughly the manuscript. Based on their comments, this manuscript requires a major revision. Some of the reasons for these decisions are given by the reviewers and I agree with them in terms of the recent literature citing and English language and spell check throughout the manuscript. Some aspects of the manuscript were vague and lengthy and may not attract a wide range of readers. Please make it concise and approach it from the drug discovery point of view so that it can be useful to young researchers interested in this field.

·

Basic reporting

The language used is good and unambiguous.Introduction is well written.Most of the references are up to date.However few references were not quoted from the recently published literature. Structure confirms to Peer review standards.Review is of broad and cross-disciplinary interest and within the scope of the journal.However quite good number of reviews are available in the past on the related topic.The authors have emphasized on the gaps which are existent on the topic entitled "Hypoxia-induced factor" and its role in liver fibrosis.

Experimental design

This review gave insight on how Hypoxia-induced factor causes hepatic fibrinogenesis and how it can act as a therapeutic target.Over all the authors have covered the literature published on the topic entitled " Hypoxia-induced factor and its role in liver fibrosis for the past 30 years( 1989-2020) in a systematic way".
The authors have addressed the review under the following headings
(A.)Hypoxia and hypoxia-inducible factors,
(B.)HIFs regulation in an oxygen-dependent manner,
(C.)HIFs regulation independent of oxygen,
(D.)Expression of HIFs in the liver during hepatic injury,
(E.)Profibrotic Function of HIF in Hepatocytes,
(F.)HIF role in liverfibrosis development,
(G.) Hypoxic Kupffer Cells and liverfibrosis,
(H.)Hepatic Stellate Cells and HIF,
(I.)The perspective of HIF as a therapeutic targetfor liverfibrosis.
(J.) Conclusion.
The authors have indicated that the following aspects of Inclusion and exclusion Criteria.
Inclusion Criteria:-.
• Only those studies were included that discussed the role of hypoxia and hypoxia-inducible factors in liver fibrosis development
• Studies from 2016 to 2021
• Those articles that discussed the potential of hypoxia-inducible factors in liver fibrosis treatment
Exclusion Criteria.
• Case studies, letters, and reports were excluded
• Studies other than English language were excluded
• Studies published before 2016-This is not followed by the authors as there are many references which were quoted prior to 2016.
• Studies that reported duplicated data.

Validity of the findings

(1.)This review dealt on the effects of Hypoxia-induced factor"(HIF) on promising liver cell types that lead to liver fibrosis.
(2.)The authors are of the opinion that information gathered from previous literature would be helpful for researchers to understand better the dynamic function of HIF especially in liver fibrosis and also to establish treatment interventions to regulate HIFs.
(3) HIF controllers are currently under development and tested in clinical trials for the treatment of ischemia and cancer.The authors conclude that some of these molecules recently underwent preclinical studies for liver transplantation.
(4.) The authors recommend HCC, translational research should be be prioritized in the coming years.

Additional comments

However the recent references published have been not been cited and discussed..
{1}.Hypoxia-Inducible Factors and Liver Fibrosis *Cells. 2021 Jul; 10(7): 1764.
Published online 2021 Jul 13. doi: 10.3390/cells10071764.PMCID: PMC8305108.PMID: 34359934
Authors :-Beatrice Foglia,† Erica Novo,† Francesca Protopapa, Marina Maggiora, Claudia Bocca, Stefania Cannito, and Maurizio Parola.
{2.}Hypoxia -induciable factors as key players in the pathogenesis of non-alcohol fatty liver disease and non-alcoholic steatohepatatis. Frontiers in Medicine and Gastroentreology, 06,,2021
Authors :-Lorenz M.W,Holzner and Andrew J.Murray.
{3.}Regulaory mecahnisms of HIF-1 alpha and its role in liver diseases: a narrative review.Annals of translational medicine. Vol.10.No.2(2022) doi.21037/atm 21-422
Authors :-Qingfeichu ,Xinyu Gu,Qinxian Zheng,Halhong Zhu.
{4.}Hypoxia-inducible factors and innate immunity in liver cancer.Vincent Wai-Hin Yuen, Carmen Chak-Lui Wong.J Clin Invest. 2020;130(10):5052-5062. https://doi.org/10.1172/JCI137553.
(5.)The Role of Hypoxia-Inducible Factor Post-Translational Modifications in Regulating Its Localisation, Stability, and Activity Adam Albanese,1 Leonard A. Daly,2 Daniela Mennerich,3 Thomas Kietzmann,3 and Violaine Int J Mol Sci. 2021 Jan; 22(1): 268.Published online 2020 Dec 29,Doi10.3390/ijms22010268.PMCID: PMC7796330.PMID: 33383924.
The authors have to consider the above references and discuss the findings accordingly.

·

Basic reporting

The review is broad and cross disciplinary interest within the scope of the journal. But, several reviews similar to this have been published previously. Therefore, I suggest the authors to incorporate a section on the existing gaps in the literature pertaining to the role of HIFs in Liver Fibrosis; and discuss on the available literature addressing these gaps. In addition, a section on future directions is needed to further strengthen this review

Experimental design

Authors should mention why they have restricted to the literature published between 2016 to date. Some of the sections, especially, the basic details of HIFs is discussed extensively. Sections like this can be shortened

Validity of the findings

The goals set out for this review are partly addressed. Conclusions and future directions sections have to be developed in a better way

Additional comments

The review article “Hypoxia-induced factor and its role in liver fibrosis” by Omar, JM et al discussed about the role of HIFs in liver fibrosis.

Comments: Authors have thoroughly discussed the molecular mechanisms that are mediated by HIF1 and the involvement of HIF1 in liver fibrosis. Even though the review is interesting, in its current form it is very exhaustive and discusses various basic aspects, which can be shortened. More emphasis can be given to strategies that are available to modulate HIF1 and its target genes. In addition, the current status of targeting HIF1 pathways can be emphasized more. Furthermore, emphasis can be given more to combination therapies involving HIF1 in fibrosis.

Inclusion criteria: Authors should provide a specific reason for selecting the data collection duration from 2016 to 2021. Even though authors mentioned that they have considered the publications published between 2016 to 2021, many of the references were dated even before 2016.

In its current format, it appears like a book chapter. Sections on Hypoxia and HIF (Lines 148 to 189); HIFs regulation – Oxygen dependent (Lines 190 to 225) and Oxygen independent (Lines 226 to 240) can be shortened significantly

Figures 2 & 3: Correct “Metastatsis” in to “Metastasis”.

It is also important that authors pay more attention and highlight how different is this review with already published review articles (Please some examples listed below). Ideally, it is better if authors can provide a section on the gaps in the existing knowledge and strategies that can be considered for addressing these gaps

Review Article | Open Access
Volume 2016 |Article ID 7629724 | https://doi.org/10.1155/2016/7629724
Clinical Advancements in the Targeted Therapies against Liver Fibrosis
Ruchi Bansal Beata Nagórniewicz,1 and Jai Prakash1

J Mol Med (Berl). 2016; 94: 613–627.
Published online 2016 Apr 20. doi: 10.1007/s00109-016-1408-1
Hypoxia-inducible factors as molecular targets for liver diseases
Cynthia Ju, Sean P. Colgan, and Holger K. Eltzschig

J Cell Biochem. 2019 Sep;120(9):14735-14744.
Hypoxia inducible factor-1 promotes liver fibrosis in nonalcoholic fatty liver disease by activating PTEN/p65 signaling pathway
Jie Han 1 2 , Yaping He 3 , Hui Zhao 2 , Xiaowei Xu 2

Biochimie Volume 108, January 2015, Pages 1-7
Review: Hypoxia-inducible factor-1alpha in hepatic fibrosis: A promising therapeutic target LeiZhanabChengHuangabXiao-MingMengabYangSongabXiao QinWuabYangYangabJunLiab

Reviewer 3 ·

Basic reporting

This paper is interesting and comprehensive. But its organization should be improved and revised to be more concise. I have some comments as follows.

 Indeed, there are lots of studies regarding HIFs in cancer. An overview of findings from meta-analyses has been published (https://amj.amegroups.com/article/view/3793). This topic should be mentioned when the authors introduced the HIF.

 The authors should greatly improve the language and grammar in this article. For example, in the "Abstract" section (line 47 on page 9), the words "play role" are not correct, and should be corrected to "play a role". In the section "HIFs regulation in an oxygen-dependent manner" (line 192 on page 18), the words "is belonged to" are wrong, and should be modified to "belongs to".

 The author should also improve the typography of this article.

 In the "Introduction" section (line 106 on page 13), "HIF-2β" is interpreted as both "ARNT 2" and "ARNT 3".

 In the section "HIFs regulation in an oxygen-dependent manner" (line 219 on page 19), the words "which is approximately 8%" are ambiguous and should be amended.

 In "Expression of HIFs in the liver during hepatic injury" section, the authors should clearly state which acute and chronic injuries are. Is "treatment of mice with ethanol, acetaminophen, and carbon tetrachloride" an acute injury or a chronic injury? (lines 258-259 on page 22)

 In the "Hepatic Stellate Cells and HIF" section, when the authors described the results of experiments, the description of the control group should be added, so that the effect of HIF-1α on liver fibrosis under in vivo conditions of liver injury is more obvious (lines 559-571 on page 42).

 In the "Hepatic Stellate Cells and HIF" section, the authors mention that in experiments with HIF-1α knockout mice, the observed results in HIF-1α knockout mice are different from those observed in in vitro studies, but the paper does not explicitly state the HIF-1α knockout results of in vitro experiments in mice. They should be revised (line 572 on page 42).

 In the "Profibrotic Function of HIF in Hepatocytes" section, the expressions on the effect of ethanol and carbon tetrachloride treatment on HIF-1α-deficient mice on liver fibrosis are unclear, and should be corrected (lines 321-320 on page 25-26).

 A previous hypothesis about the role of HIF in Budd-Chiari syndrome should be mentioned (PMID: 29984199).

 The originality of figures should be confirmed.

Experimental design

See the comments above.

Validity of the findings

See the comments above.

---

## Round 0.2 · Minor Revisions

Please respond to reviewer #3 comments and incorporate them

·

Basic reporting

The Authors have attended to all the suggestions of the three reviewers and modified the manuscript accordingly.

Experimental design

The authors have incorporated the corrections as suggested by reviewers 1,2 and 3.

Validity of the findings

The authors have incorporated the findings from the recent references which have been published from 2016-2021 ( Reviewer 1).The recent references published have not been cited and discussed.
{1}.Hypoxia-Inducible Factors and Liver Fibrosis *Cells. 2021 Jul; 10(7): 1764.
Published online 2021 Jul 13. doi: 10.3390/cells10071764.PMCID: PMC8305108.PMID: 34359934 Authors :-Beatrice Foglia,† Erica Novo,† Francesca Protopapa, Marina Maggiora, Claudia Bocca, Stefania Cannito, and Maurizio Parola.
{2.}Hypoxia -induciable factors as key players in the pathogenesis of non-alcohol fatty liver disease and non-alcoholic steatohepatatis. Frontiers in Medicine and Gastroentreology, 06,,2021 Authors :-Lorenz M.W,Holzner and Andrew J.Murray.
{3.}Regulaory mecahnisms of HIF-1 alpha and its role in liver diseases: a narrative review.Annals of translational medicine. Vol.10.No.2(2022) doi.21037/atm 21-422
Authors :-Qingfeichu ,Xinyu Gu,Qinxian Zheng,Halhong Zhu.
{4.}Hypoxia-inducible factors and innate immunity in liver cancer.Vincent Wai-Hin Yuen, Carmen Chak-Lui Wong.J Clin Invest. 2020;130(10):5052-5062. https://doi.org/10.1172/JCI137553.
(5.)The Role of Hypoxia-Inducible Factor Post-Translational Modifications in Regulating Its Localisation, Stability, and Activity Adam Albanese,1 Leonard A. Daly,2 Daniela Mennerich,3 Thomas Kietzmann,3 and Violaine Int J Mol Sci. 2021 Jan; 22(1): 268.Published online 2020 Dec 29,Doi10.3390/ijms22010268.PMCID: PMC7796330.PMID: 33383924.
The authors must consider the above references and discuss the findings accordingly.
Noted, addressed, and cited all the mentioned references
P ~ 4, line 118-128,
P ~12,13, line 397-432

Additional comments

Nil.

·

Basic reporting

Basic reporting is clear and professionally performed. The figures are appropriate and discussed

Experimental design

Appropriate. Authors have performed rigorous review of the literature and discussed

Validity of the findings

Conclusions are well reported and highlight existing gaps in the literature

Additional comments

None

Reviewer 3 ·

Basic reporting

The authors have revised the questions raised last time, but the authors are still too careless and have made many mistakes.

1. The writing of the abbreviation "Hif" appears many times in the article.
2. The authors should standardize the writing of "hypoxia inducible factor". For example, when reviewing this article, it can be seen that the author uses "HIF1" and "HIF-1" at the same time. The authors are requested to revise it carefully.
3. The authors mistakenly wrote the Greek letter into the English letter many times.For example, the authors mistakenly wrote "α" as "a" and "β" as "b".
4. Some references are mistakenly cited in the article, such as page 18 on line 404.
5. When the abbreviations of proper nouns appear in the article, they should be explained. For example, in line 96 on page 8, the authors should clearly indicate the meaning of "FAO".
6. The authors are too careless, and some contents in pages 9 and 10 are completely the same.
7. In the section of "Profibrotic Function of HIF in Hepatocytes", the authors said "HIF-2α knockout even though prevented the enhanced progression of fibrosis, it also culminated in overall decreased liver damage". The expression should be improved.
8. The authors should add a section introducing the illustrations in the article so that the readers can better understand the illustrations.
9. Are some figures obtained from other published papers?

Experimental design

This is a review.

Validity of the findings

This is a review.

---

## Round 0.3 · accepted · Accept

All comments from the previous version are attended to and with satisfactory responses. This paper can be accepted for publication as is.